# Proline Spray Relieves the Adverse Effects of Drought on Wheat Flag Leaf Function

**DOI:** 10.3390/plants13070957

**Published:** 2024-03-26

**Authors:** Huizhen Li, Yuan Liu, Bo Zhen, Mouchao Lv, Xinguo Zhou, Beibei Yong, Qinglin Niu, Shenjiao Yang

**Affiliations:** 1Institute of Farmland Irrigation, Chinese Academy of Agricultural Sciences, Xinxiang 453002, China; lihuizhen@caas.cn (H.L.); zhouxinguo@caas.cn (X.Z.); niuqinglin@caas.cn (Q.N.); yangshenjiao@caas.cn (S.Y.); 2Graduate School of Chinese Academy of Agricultural Sciences, Beijing 100091, China; 3Jiangsu Vocational College of Agriculture & Forestry, Jurong 212499, China; 18452651905@163.com

**Keywords:** antioxidant enzyme, anatomic feature, plant growth regulators, water deficit, crop

## Abstract

Drought stress is one of the key factors restricting crop yield. The beneficial effects of exogenous proline on crop growth under drought stress have been demonstrated in maize, rice, and other crops. However, little is known about its effects on wheat under drought stress. Especially, the water-holding capacity of leaves were overlooked in most studies. Therefore, a barrel experiment was conducted with wheat at two drought levels (severe drought: 45% field capacity, mild drought: 60% field capacity), and three proline-spraying levels (0 mM, 25 mM, and 50 mM). Meanwhile, a control with no stress and no proline application was set. The anatomical features, water-holding capacity, antioxidant capacity, and proline content of flag leaves as well as grain yields were measured. The results showed that drought stress increased the activity of catalase and peroxidase and the content of proline in flag leaves, lessened the content of chlorophyll, deformed leaf veins, and decreased the grain yield. Exogenous proline could regulate the osmotic-regulation substance content, chlorophyll content, antioxidant enzyme activity, water-holding capacity, and tissue structure of wheat flag leaves under drought stress, ultimately alleviating the impact of drought stress on wheat yield. The application of proline (25 mM and 50 mM) increased the yield by 2.88% and 10.81% under mild drought and 33.90% and 52.88% under severe drought compared to wheat without proline spray, respectively.

## 1. Introduction

Wheat (*Triticum aestivum* L.) is the most important staple crop in temperate zones, which feeds 40% of the world’s population [1]. In China, it is mainly grown in the Yellow and Huai River valleys, the middle and lower reaches of the Yangtze River, the Southwest China, the Loess Plateau, the Northeast China, and the Xinjiang province [2]; the average yield and the total production in 2023 is 5.83 t/ha and 134.53 million tons, respectively [3]. The synchronization of crop phenological development stages with environmental conditions is crucial for yield production [4], and water is an important environmental condition. In recent years, the frequency, intensity, and duration of drought events have increased, threatening agricultural productivity and food security [5]. Drought stress has consistent negative effects on yield and yield components, but does not always decrease water-use efficiency for grain [6]. Soil drought leads to the closure of plant stomata, a decrease in chlorophyll content [7] and photosynthetic rate [8], the disorder of enzyme activity, and a decline in CO_2_ intake [9,10], resulting in irreversible harm to plants [11]. It is reported that drought stress (30% field capacity) can reduce total biomass production and root–shoot ratio [12]. Hameed et al. reported that compared to the treatment with 100% of total available soil water, the activities of catalase (CAT) and peroxidase (POD) in wheat leaves increased when the total available soil water was at 50%, while the activity of superoxide dismutase (SOD) either increased or remained unaffected [13]. Sudden drought stress and a permanent water deficit exert negative effects on the anatomic structure of flag leaves of wheat [14]. The heading and anthesis periods are pivotal for wheat yield formation, and they are extremely susceptible to drought due to its high evapotranspiration. Drought during this period will lead to a decrease in the number of grains per spike and thereby yield. The moderate drought during the anthesis period, with 50% field capacity, can lead to a yield reduction ranging from 19% to 42% compared with the treatment of being well-watered [15].

Under drought stress, plants can ensure cellular integrity and adjust the accumulation of substances through osmotic regulation [16]. Osmoregulatory substances include several inorganic solutes such as K^+^, Ca^2+^, silicon, and salicylic acid, as well as organic solutes including glycine, betaine, proline, soluble carbohydrates, and proteins [17,18,19,20]. The osmotic regulation under drought stress has beneficial impacts on the yield of most crops [21]. Morgan reported that the flag leaves of wheat exhibited osmoregulation under the 28-d drought stress including 14 days before anthesis and 14 days after anthesis [22]. From then on, flag leaves play a crucial role in the research of osmoregulation under drought-stress conditions [23,24].

The application of exogenous osmoregulatory substances under abiotic stress conditions can enhance the accumulation of osmotic substances and antioxidants, thereby maintaining the balance of osmotic pressure and eliminating reactive oxygen species, and ensuring the stability of membrane structure, enzymes, and other macromolecules. This enables plants to respond more rapidly and effectively to stress [25]. Therefore, in recent years, researchers have attempted to apply diverse plant growth regulators (PGRs), such as osmoregulation substances [26], to enhance crop drought resistance. PGRs have emerged as a forefront of research due to their practicality and environmental friendliness [27]. Proline is a widely used osmoregulatory substance that exists in higher plants, always engaged in drought response through osmoregulatory mechanisms. Its primary function is to safeguard cellular processes and organs [28]. Kemble and Macpherson first observed the accumulation of proline in perennial ryegrass (*Lolium perenne*) after wilting [29].

The techniques of applying proline mainly include seed soaking and foliar spraying. Previous studies have found that the exogenous application of proline (30 mM) can effectively alleviate the adverse impacts of osmotic stress of 100 mM NaCl on rice-seeding growth (*Oryza sativa*) [30]. Hussain et al. [31] reported that the foliar applications of proline (2.5 mM) can effectively improve water-use efficiency and the enzymatic activities of *Abelmoschus esculentus* L. under high-temperature stress. Ali et al. reported that foliar-spraying proline (30 mM) at seedling and/or vegetative stages of maize (*Zea mays*) can facilitate maize growth by improving photosynthesis and increasing antioxidant compounds under 60% field capacity [32], and promote the absorption of K^+^, Ca^2+^, N, and P by maize, thus counteracting the negative effects of drought stress on nutrient absorption [33]. Soaking seeds with proline (20 mM) before sowing relaxed the bad impacts of drought stress (60% field capacity) on wheat and increased wheat yield [34]. Molla et al. [35] found that both proline and betaine exert a protective function in the oxidative stress response induced by drought, as they mitigate H_2_O_2_ levels and enhance antioxidant defense systems, but proline exhibited a superior efficacy compared to betaine.

However, it remains unresolved whether exogenously applied proline during the heading and anthesis stages can ease the detrimental effects of drought stress on wheat growth. We hypothesized that spraying proline could alleviate the adverse effects of drought stress on wheat growth, and that the effectiveness would vary with spraying concentration and severity of drought, since exogenous proline has the potential to regulate chlorophyll content, antioxidant enzyme activity, osmotic regulation substance content, and the leaf tissue structure of plants under abiotic stress. Therefore, we conducted a barrel experiment with varying degrees of water scarcity and proline spray concentrations and measured the physiological and biochemical indices of wheat flag leaves as well as yield to investigate the alleviating effect and mechanism of proline on wheat under drought stress. The results can provide support for sustainable agricultural development under water-limited conditions.

## 2. Results

### 2.1. Drought Lowers Evapotranspiration

The evapotranspiration rate was decreased by drought, and the reduction range was consistent with the degree of drought (Figure 1). Especially, the evapotranspiration rate under severe drought without proline spraying (H0) was significantly decreased by 72.59% compared with CK. This can be attributed to the closure of stomata and the reduction in leaf transpiration caused by drought stress. Exogenous proline application significantly decreased evapotranspiration compared to the treatment subjected to drought without proline spray, and there was no significant difference between different proline concentrations.

### 2.2. Proline Spraying Promotes Water-Holding Capacity and SPAD of Wheat Flag Leaves

The results of two-way ANOVA showed that the degree of soil drought and concentration of proline had a significant effect on the saturated water content of isolated flag leaves of wheat during the flowering and seed-formation stages (Table 1). The water-holding capacity of flag leaves (saturated water content (SWC) and relative water-content change rate (RWC)) is illustrated in Figure 2. As shown in Figure 2a, there was no statistically significant difference in SWC of flag leaves between drought treatments and CK at the anthesis stage. However, proline spray significantly increased the SWC of flag leaves compared to the treatment of drought without proline spraying, except for the treatment of severe drought sprayed with 25 mM proline. At the seed-formation stage, the change rate of relative water content in the treatment of drought and proline spray increased significantly compared with CK (Figure 2c).

After 7 days of rehydration, at the seed-formation stage, drought stress decreased the SWC of flag leaves compared with CK, and severe drought stress decreased the SWC significantly compared with treatment under mild drought conditions (Figure 2b). Proline spray increased the SWC compared with drought-stressed treatments, among which the treatment with a spraying concentration of 50 mM boosted significantly. The relative water-content change rate of M0 and H0 also decreased compared with CK, with the greater reduction in H0 (Figure 2d). The RWC loss rate of wheat leaves under drought stress decreased initially and then increased with the rise in proline application concentration.

The degree of soil drought had a significant effect on the SPAD value of flag leaves of wheat during the seed-formation stage (two-way ANOVA, Table 2). The SPAD values of flag leaves at the heading stage ranged from 43.55 to 46.43, with no significant differences among treatments (Table 3). As the growth period progressed, the SPAD value in CK increased slowly, while the SPAD values of M0 and H0 decreased. The SPAD value of flag leaves was weakened by drought stress, especially by severe drought, and accelerated by proline spray, although this effect was not statistically significant. In most cases, different proline concentrations did not have a statistically significant influence on the SPAD values of flag leaves, except that, under mild drought, the spraying of 25 mM proline significantly increased the SPAD values at the seed-formation stage compared to the treatment without proline spray (M0).

### 2.3. Exogenous Proline Improves the Performance of Wheat Antioxidant Enzyme System

As shown in a two-way ANOVA, the degree of soil drought and the concentration of proline significantly impacted the malondialdehyde (MDA) content, superoxide dismutase (SOD), catalase (CAT), and peroxidase (POD) enzyme activities during the flowering and grain-filling stages (except for the degree of soil drought during the grain-filling stage) (Table 4). The MDA content in wheat flag leaves subjected to drought stress was higher than that of CK after anthesis drought, and the MDA content of flag leaves in mild drought (M0) was significantly increased (Figure 3a). Furthermore, the exogenous application of proline significantly increased the MDA content of wheat under drought conditions. Compared with CK, the MDA content of severe drought + 50 mM proline treatment (H50) was significantly increased by 41.40%, while that of mild drought + 25 mM proline treatment (M25) was significantly increased by 77.88%. After two weeks of rehydration (11 May 2022) at the grain-filling stage, the MDA content in wheat under drought was still higher than that of CK significantly; however, the differences among M0, H0, and CK were reduced significantly compared with those observed during anthesis stage.

The SOD activity in flag leaves of wheat under drought stress was significantly decreased compared to CK (Figure 3b), especially under severe drought conditions. Compared with treatment without proline spraying, the SOD activity was significantly increased by proline spraying. After rehydration, the SOD activity in the flag leaves of wheat subjected to drought stress but not sprayed with proline increased compared with CK, and the increase degree was consistent with the severity of drought. The SOD activity in the flag leaves of wheat sprayed with proline did not show a significant increase compared to those treated without proline at the same level of drought.

The CAT activity in the flag leaves of wheat increased significantly compared to CK before and after rehydration due to drought (Figure 3c). Compared to the treatment without proline spraying, the CAT activity in the flag leaves of wheat treated with proline spraying under mild drought conditions significantly increased after drought-stress treatment, while the CAT activity in flag leaves treated with proline spraying under severe drought conditions significantly increased after rehydration.

Before and after rehydration, the POD activity in the flag leaves of wheat increased significantly under mild drought conditions without proline spraying compared to CK (Figure 3d). However, under severe drought conditions, the activity did not increase. Compared with the treatment without proline spray, the POD activity in flag leaves of wheat under severe drought condition and sprayed with proline was significantly increased, before and after rehydration. Additionally, under mild drought conditions, proline spray significantly increased the POD activity of flag leaves after rehydration

### 2.4. Exogenous Proline Significantly Increases the Content of Proline in the Flag Leaves of Wheat

The proline content during the flowering and grain-filling stages was significantly affected by the degree of soil drought and concentration of proline, except for the soil drought degree at the grain-filling stage (Table 4). At the anthesis stage, after drought-stress treatment, there was a significant increase in proline content in wheat flag leaves under drought stress compared to CK, with the greater increase when sprayed with proline (Figure 4). At the grain-filling stage, the proline content of CK increased, and the gap between CK and drought treatment narrowed; and under mild drought conditions, the proline content in flag leaves of wheat sprayed with proline decreased than that at anthesis stage; meanwhile, there was no statistical difference in the proline contents of the flag leaves of wheats sprayed with different concentrations of proline under the same drought condition.

### 2.5. The Anatomic Characteristics of Flag Leaves Are Elevated by the Suitable Concentration of Proline Spray

The degree of soil drought and the concentration of proline both exerted a significant effect on the perimeter and area of vascular bundle of the midvein, while the thickness, perimeter, and area of midvein were only significantly influenced by the concentration of proline (Table 5). Meanwhile, the drought degree and proline concentration both did not exert a significant effect on the perimeter and area of large and small vascular bundles (Table 6). The influence of exogenous proline on the anatomic characteristics of the midvein of wheat flag leaves under drought stress is depicted in Figure 5. Compared with CK, the thickness (Figure 5a), perimeter (Figure 5b), and area (Figure 5c) of the midvein in wheat flag leaves under drought stress were significantly reduced. Compared to the treatment under identical drought conditions without proline spraying, treatments of severe drought + 50 mM proline and mild drought + 25 mM proline could significantly increase the thickness, perimeter, and area of the leaf midvein, and severe drought with a 25 mM proline application had a positive impact on the thickness of leaf midveins, whereas the treatment of mild drought + 50 mM proline resulted in a decrease in leaf midvein thickness.

Compared with CK, the perimeter and area of midvein vascular bundles (Figure 5d,g), large vascular bundles (Figure 5e,h), and the area of little vascular bundles (Figure 5i) in wheat flag leaves decreased significantly with severe drought. However, mild drought only significantly decreased the perimeter, area of large vascular bundles, and the area of small vascular bundles. The treatments of applying 50 mM proline under severe drought and 25 mM proline under mild drought augmented the perimeter and area of midvein bundles, large vascular bundles, and small vascular bundles, and the area of flag leaves compared to CK and the corresponding drought treatments without proline. The impact of severe drought with 25 mM proline application and mild drought with 50 mM proline application on the perimeter and area of all kinds of vascular bundles was minimal; however, both treatments resulted in an increase in the midvein vascular bundle area without increases in perimeter.

### 2.6. Exogenous Proline Significantly Increased Wheat Yield under Severe Drought Conditions

The yield was significantly affected by both the drought degree and proline application level, while the ear number, grain number per ear, and thousand-grain weight were only significantly impacted by the soil drought degree (Table 7). 

Both mild and severe drought at the heading and anthesis stages deteriorated the yield composition and yield per barrel of wheat compared with CK, of which the effect of severe drought was significant (Table 8). Mild drought and severe drought caused 11.72% and 48.24% of yield reduction compared with CK, respectively. Under the mild drought condition, the exogenous spraying of proline (25 and 50 mM) increased ear number, grain number per ear, and yield of wheat, with the most beneficial effect in 25 mM. 

Under the severe drought condition, exogenous spraying also increased the yield composition and yield of wheat compared to the treatment without proline spraying. Relative to H0, the yield increase for each barrel in H50 (52.38%) was higher than that in H25 (33.33%).

## 3. Discussion

### 3.1. Effects of Exogenous Proline on the Function of Flag Leaves of Wheat under Drought Stress

In this study, the application of exogenous proline increased the saturated water content in the flag leaves of stress-affected wheat at the end of drought treatment (the anthesis stage), because proline application can regulate the osmotic regulation of cells, enhancing their water-capture and -absorption ability [36]. After rehydration, the proline-sprayed wheat leaves had a higher saturated water content, while the drought-treated wheat leaves had a lower saturated water content. The spraying of proline possibly reduced transpiration, xylem tension, and cavitation caused by water-column breakage in xylem, and better maintained the shape and size of leaf veins. Consequently, the xylem in the veins of leaves was not prone to embolization [37], and ensured the transport and storage of water [38], which was conducive to improving the water absorption and retention of leaves, as well as the drought resistance of plants [39].

Applying exogenous proline with appropriate concentrations enhanced the water-holding capacity of flag leaves under drought stress, which was consistent with previous studies’ conclusions that the accumulation of proline in maize root tip cells was beneficial to cell expansion and water absorption [40]. However, the application of exogenous proline at 50·mM increased the relative water-content loss rate of drought-affected wheat after rehydration compared with that at 25 mM, which may be because the accumulation of reactive oxygen species caused by the high concentration of proline resulting in the premature cell aging [41].

An interesting finding of this study was that in the process of drought control, evapotranspiration in the treatment of spraying with proline was significantly lower than that in the corresponding treatment without proline, and the difference between the proline treatments with varying concentrations was not significant. These phenomena indicated that proline application inhibited the total evapotranspiration of the barrel including the wheat and the soil. Since the proline solution was sprayed evenly onto the wheat plants, it was inevitably sprayed also onto the bare soil surface. In future, the impact of proline spraying on soil-surface evaporation and on leaf-surface transpiration needs to be distinguished.

### 3.2. Effects of Exogenous Proline on Antioxidant System of Flag Leaves of Wheat under Drought Stress

In our study, the content of MDA in the flag leaves of wheat was elevated under drought stress, as MDA content typically increases in plants experiencing stressful conditions and it is an indicator of membrane impairment [42]. Another intriguing observation in this study is that exogenous proline further increased the MDA content in the flag leaves of wheat. This is different from the previously reported results that exogenous proline decreased [43,44] or did not affect MDA content [45] in plants. In general, proline application could reduce MDA content in plants. But the wheat variety in this study appeared to be sensitive to proline. The toxicity of excessive proline to plants such as an increase in reactive oxygen species has been reviewed [41], and it is also reported that the increase in reactive oxygen species increased MDA content in the leaves of grapevine [46]. Accordingly, MDA content increased after proline spraying and decreased after rehydration in this study. In this study, exogenous proline lowered the MDA content in the flag leaves of wheat after rehydration, which was in line with the literature [43].

Here, severe drought stress at the heading and anthesis stages inhibited the SOD and POD activities of wheat flag leaves, while mild drought did the opposite. In a separate study, under 60% field water capacity, the antioxidant enzyme activities of cowpea (*Vigna unguiculata*) leaves were significantly increased [47], which is a defensive mechanism of plants in adverse environments. We found that the activities of antioxidant enzymes in flag leaves subjected to drought degrees were recovered in most cases by spraying proline, consistent with two tobacco experiments [48,49].

Also, drought stress promoted the increase in proline content in the flag leaves of wheat, and the increased amplitude was positively correlated with the degree of drought in this study. Considering that proline plays a pivotal role in safeguarding cellular and organ integrity [28], and its metabolism can stabilize cell homeostasis under stress, it is reasonable that the proline spray on the leaf surfaces augmented the proline content inside the wheat flag leaves, although they showed a decreasing trend under mild drought conditions after rehydration in our study. The decrease in proline content in plants after rehydration was due to relief of stress, thus the down-regulation of proline biosynthesis pathway enzymes and the up-regulation of proline degrading enzymes [50].

### 3.3. Effects of Exogenous Proline on Anatomical Characteristics of Flag Leaves of Wheat under Drought Stress

We observed that the anatomical dimensions of the midvein in wheat flag leaves were obviously redacted under drought stress during the heading and anthesis stages. It was consistent with Ghafoor’s report that leaf midvein thickness and vascular bundle area dropped sharply under a 60% field water capacity [51]. This may be attributed to the inhibitory effect of drought on cell expansion [52]. Furthermore, from a functional perspective, the sclerified tissues of the midvein provide flexural rigidity to avoid bending [53]. According to the cantilever-beam model [54], the leaf planar dimensions decreased under drought stress, requiring less flexural rigidity [52,54]. Therefore, the deterioration of the flag leaf midvein function according to the use and disuse theory results in tissue atrophy. In this study, the midvein perimeter of the flag leaves of mild and severe-drought-stressed wheat was almost equal, while the midvein area of the flag leaves of mild drought wheat was larger, indicating a greater destruction caused by severe drought. The perimeter and area of large, small, and midvein vascular bundles evidently decreased due to severe drought. This phenomenon occurs due to plants’ ability to decrease the size of vascular bundles to uphold water potential and cell turgor pressure under drought stress [55,56]. Moreover, the narrower vascular are less prone to cavitation, which mitigate the occurrence of vein embolism [57] and advance the adaptation of wheat to drought.

In the current study, we observed that the proper concentration of proline was beneficial to preserving the anatomical characteristics of wheat flag leaves. Under severe and mild drought conditions, 50 mM and 25 mM of exogenous proline performed better, respectively. This may be attributed to the accumulation of proline in the leaves, which helps to maintain the cell turgor pressure and morphology of the cells. These findings are consistent with previous research on oat (*Avena sativa* L.) at 60% field water capacity where exogenous proline (40 mM) had a similar effect [51]. However, under mild drought conditions, the anatomical characteristics increased first and then decreased with increasing concentrations of applied proline. This suggests that 50 mM of proline may be too much for mild drought conditions because of its potential toxicity.

### 3.4. Effects of Exogenous Proline on Wheat Yield under Drought Stress

Drought apparently decelerated the number of wheat ears, and severe drought significantly decreased the number of grains per ear in our study. At the sensitive anthesis stage, drought can directly affect the number of grains and lead to a notable decrease in grain yield [58,59]. Under drought-stress conditions, the number of ears per plant and grains per ear as well as the 1000-grain weight were improved by proline spray (Table 2). This is because spraying proline resulted in the accumulation of proline in the flag leaves (Figure 4), enhancing their osmotic regulation ability and maintaining the cell morphology and tissue integrity of the leaves (Figure 5). This contributed to the maintenance of leaf water-holding capacity (Figure 2), the increase in antioxidant enzyme activity, and the reduction of lipid peroxidation after rehydration (Figure 3), thereby retarding leaf aging. The elevation in leaf chlorophyll content (Table 1) facilitated photosynthetic recovery, ultimately mitigating the adverse effects of drought stress on wheat grain yield. It is in agreement with the results that 150 mg/L (1.3 mM) proline spray effectively increased the yield of wheat under 35% field water capacity [60]. Similarly, exogenous proline enhanced wheat [61] and maize [62] biomass and ultimately grain yield under high-salt conditions.

## 4. Materials and Methods

### 4.1. Experiment Site

This experiment was carried out in the comprehensive experimental station of Xinxiang, Chinese Academy of Agricultural Sciences (35°19′ N, 113°53′ E, altitude 73.2 m) from October 2021 to May 2022. The station, located in Xinxiang, Henan Province, China, experiences a temperate continental monsoon climate, and its average annual rainfall, temperature, and evaporation are recorded as 573.4 mm, 14 °C, and 2000 mm, respectively. The test soil was collected from the tilling layer (0–25 cm soil layer) of the field, which is a fluvo-aquic soil according to Chinese classification, and a Fluvic Cambisol according to the World Reference Base. The basic properties are listed in Table 9. After air-drying, breaking, and sifting, a total of 17.5 kg soil was loaded into each plastic barrel (top diameter: 31 cm, bottom diameter: 24 cm, height: 29 cm) which was buried into the field with the upper 5 cm above the soil surface to effectively restore the soil environment. The repacked soil field water capacity, permanent wilting point, and bulk weight was 26.86%, 8%, and 1.44 g/cm^3^, respectively. The fertilizer was applied at a rate of N 400 kg/ha, P 75 kg/ha (P_2_O_5_ 150 kg/ha), and K 300 kg/ha (K_2_O 360 kg/ha), equating to N 0.14 g, P 0.03 g (P_2_O_5_ 0.06 g), and K 0.05 g (K_2_O 0.06 g) per kilogram of soil.

### 4.2. Experiment Design

In this experiment, two soil water-level gradients were established: mild drought (60% field capacity) and severe drought (45% field capacity), and three proline concentrations were set up: 0 mM, 25 mM, and 50 mM, with a full factor-experimental design. Another unstressed scenario (CK: 75% field capacity) with no proline application was set as the control. In total, 7 treatments were set up, each of which used 40 barrels. Wheat (*Triticum aestivum* L., cv. Lunxuan69) seeds were sown on 23 October 2021. Seedings were thinned to 30 per barrel on 27 November 2022. The experiment was carried out during the heading and anthesis stages, which are sensitive to water deficiency.

#### 4.2.1. Soil Moisture Control

When the wheat entered the heading stage, we began to control the soil water from 17 April 2022. Considering the root depth of wheat, soil in the 0–20 cm layer was collected using a soil drill (diameter 2 cm) in each barrel, then was placed in an aluminum soil moisture-content tin and brought back to the laboratory for moisture determination using a drying method. The sifted air-dried soil in 4.1 were employed to backfill the hole caused by the drill. On the afternoon of 17 April 2022, irrigation was carried out according to the difference between the measured soil moisture content and the expected water content. The different treatments officially begun on 21 April 2022, and soil moisture was controlled as mentioned above on 21, 23, and 25 April 2022. The water treatments were finished after wheat anthesis on the evening of 27 April 2022. The actual soil mass moisture content (SMC) of different treatments on the beginning day of the drought stress treatment (21 April 2022) is shown in Figure 6. While there is a disparity between the actual and expected moisture content, a discernible gradient occurred among the control, severe drought, and mild drought conditions. From the evening of 27 April to 2 days before harvest, all the treatments were irrigated with 1 L/barrel of water every 2 or 3 days according to weather condition to fully meet the water needs of wheat growth.

#### 4.2.2. Foliar Spraying Method

Proline (purity ≥ 98.5%), obtained from Sinopharm, was utilized in this study. To enhance adhesion to wheat leaves, proline was mixed with 0.1% Tween-20 solution before spraying. The inclusion of Tween-20 aimed to counteract the hydrophobic nature of leaf surfaces, thereby facilitating the effective wetting and penetration of the proline solution into the leaves [63,64]. A volume of 4.5 L of proline solution was sprayed per treatment (40 barrels) each time using a 5-L electric watering can. During the stress period, the spraying took place 4 times (21, 23, 24, and 25 April 2022) on windless evenings.

### 4.3. Item Measurement

#### 4.3.1. Evapotranspiration

Evapotranspiration was calculated as follows:
(1)$${ET}_{a}=\frac{(W_{S}-W_{S}^{\prime })\times M_{S}÷\rho +V}{A\times T}\times 1000$$

The *ET_a_* refers to the evapotranspiration (mm/d), *W_S_* refers to the soil mass moisture content measured by a drying method, $W_{S}^{\prime }$ refers to the soil mass moisture content measured by a drying method before the last irrigation, *M_S_* refers to weight of air-drying soil in the barrel; here, it is 17.5 kg. *V* refers to the irrigation amount of the last irrigation, *ρ* refers to the density of water (kg/m^3^), *A* refers to the area of soil in the barrel (m^2^), and *T* refers to the time interval between two measurements of soil mass moisture content (d).

#### 4.3.2. Leaf Mass Moisture Content

For each treatment, at the anthesis stage (27 April 2022, BBCH 69) and seed-formation stages (4 May 2022, BBCH 71), 5 wheat flag-leaf samples with similar growth potentials from 3 randomly selected barrels were cut off with leaf sheaths, placed in clean water, and brought back to the laboratory [65]. The sampled barrels were discarded each time to minimize the sampling effect on the leaf growth. After standing for 4–5 h, the leaf sheaths were cut off and the remains were weighed as m_0_. After hanging and standing for 24 h, they were weighed again as m_1_. Then, they were put into the oven at 85 °C and baked to a constant weight recorded as m_d_. The saturated and relative water content as well as the relative water-content change rate were calculated as follows:
(2)$$\mathrm{Saturated}\;\mathrm{water}\;\mathrm{content}\;(\%)\;(\mathrm{SWC})\;=\;(m_{0}\;-\;m_{d})/m_{d}\;\times \;100$$
(3)$$\mathrm{Relative}\;\mathrm{water}\;\mathrm{content}\;(\%)\;(\mathrm{RWC})\;=\;(m_{1}\;-\;m_{d})/(m_{0}\;-\;m_{d})\;\times \;100$$(4)$$\mathrm{Relative}\;\mathrm{water}\;\mathrm{content}\;\mathrm{change}\;\mathrm{rate}\;=\;(m_{1}\;-\;m_{d})/(m_{0}\;-\;m_{d})/24$$

#### 4.3.3. Chlorophyll Measurement

The chlorophyll content of five flag leaves in each barrel from three selected barrels was measured using a handheld SPAD measuring instrument at noon under sunny conditions, specifically during the heading stage (BBCH 49), anthesis stage, and seed-formation stage of wheat.

#### 4.3.4. Antioxidant-System Determination

Each treatment included the anthesis stage (27 April 2022, BBCH 69) and the grain-filling stage (11 May 2022, BBCH 75). For each barrel, five leaves were carefully excised, wrapped in tin foil, rapidly frozen in liquid nitrogen, and transported to the laboratory for further analysis. The activity of superoxide dismutase (SOD), catalase (CAT), leaf peroxidase (POD), and proline content were determined through enzyme-linked assays using an ELISA detection kit assisted by an enzyme-labeled instrument Tecan Infinite F50 (Tecan, Männedorf, Switzerland).

#### 4.3.5. Anatomical Feature Qualification

On 27 April 2022, three 10 mm fresh subsamples were collected from the middle section of flag leaves for each treatment. After immersion, the specimens were embedded in paraffin. Subsequently, they were sliced, dehydrated using xylene and anhydrous ethanol, and stained with the saffrane-solid green method. To ensure preservation, neutral gum was applied to seal the sections. Then, NIKON DS-U3 (Tokyo, Japan) was used to capture images under an overhead microscope (NIKON ECLIPSE E100 (Tokyo, Japan)), enabling the measurement of the thickness, circumference, and area of leaf midveins as well as large- and small-vascular-bundle tissues.

#### 4.3.6. Grain Yield Characterization

After harvest on 25 May 2022, five barrels of wheat were selected for each treatment and transported to the laboratory for ear and grain-number-per-ear counting, as well as the calculation of the 1000-grain weight and yield per barrel after grain drying.

### 4.4. Data Analysis

The data were analyzed using SAS 9.4 and presented as the mean ± standard error. A statistical comparison of plant properties between treatments was assessed by the Duncan test with the significance level at *p* < 0.05. The significance of the drought stress degree and proline concentration on monitored characteristics were analyzed by a two-way ANOVA using SAS 9.4. The results of the data analysis were plotted using Excel2019.

## 5. Conclusions

Drought stress, particularly severe drought, significantly impacted the growth and physiological function of wheat flag leaves, including leaf saturated water content, water-holding capacity, SPAD value, and leaf anatomical characteristics. Ultimately, it leads to a reduction in wheat yield. Luckily, the application of an appropriate concentration of exogenous proline could recover the adverse drought-affected performance of wheat flag leaves. These improvements accordingly resulted in increased water-holding capacity and wheat ear number, and especially improved the 1000-grain weight, leading to an overall increase in the grain yield of drought-affected wheat. Based on this study, we propose that 25 mM of proline is appropriate for mild drought conditions, while 50 mM is suitable for severe drought conditions to alleviate the adverse effect caused by drought stress. Based on this study, we believe that exogenous proline has the potential to cope with drought stress in field-crop production in the future.

## Figures and Tables

**Figure 1 plants-13-00957-f001:**
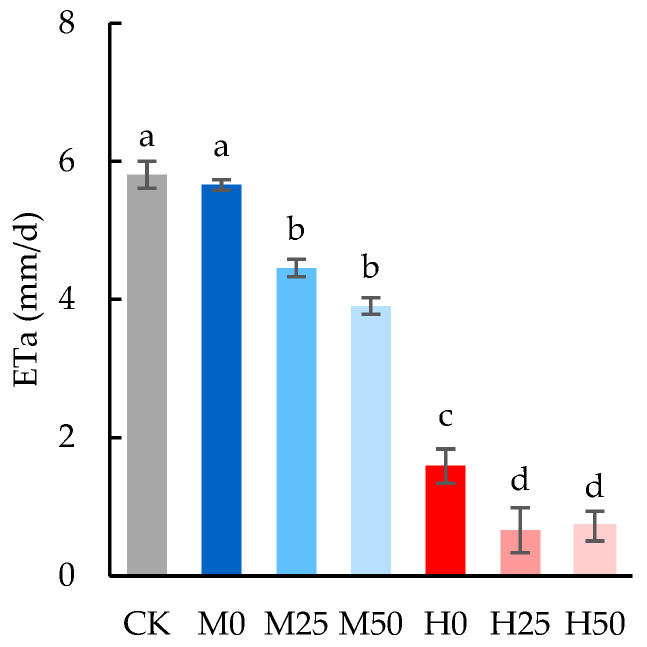
The evapotranspiration (ETa) rate during drought stress. The data are expressed as the mean ± standard error (*n* = 3); different lower-case letters above each column show significance difference among treatments. CK refers to no drought stress and no proline application, M refers to 60% field water capacity, H refers to 45% field water capacity; 0, 25 and 50 denote 0, 25 mM, and 50 mM of proline solution applied in this study, respectively.

**Figure 2 plants-13-00957-f002:**
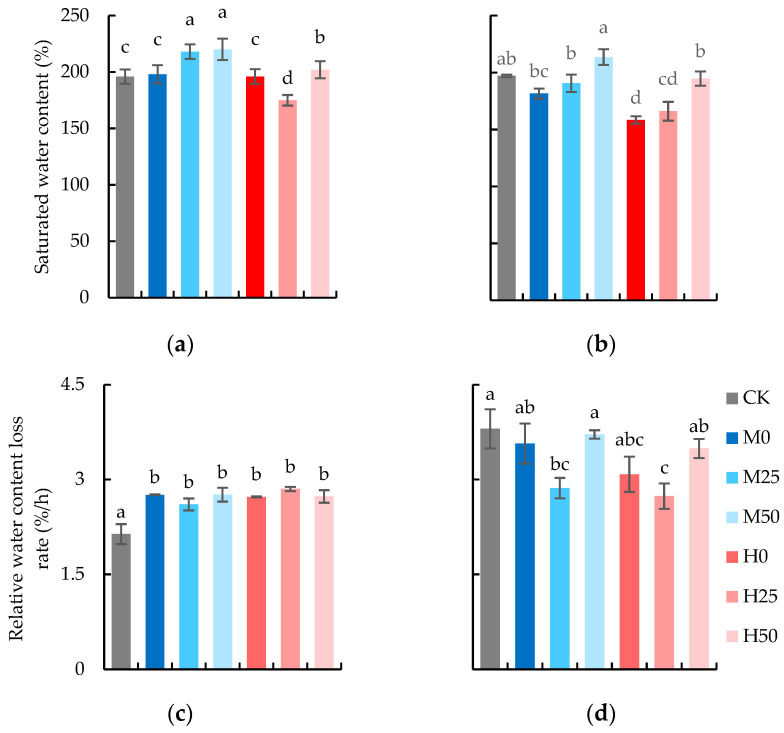
Water-holding capacity of isolated flag leaves: saturated water content at anthesis stage (**a**) and seed-formation stage (**b**), relative water-content loss rate in 24 h at anthesis stage (**c**) and grain-seed-formation stage (**d**). The data are expressed as the mean ± standard error (*n* = 3); different lower-case letters above each column show significance difference among treatments. CK refers to no drought stress and no proline application, M refers to 60% field water capacity, H refers to 45% field water capacity; 0, 25, and 50 denote 0, 25 mM, and 50 mM of proline solution applied in this study, respectively.

**Figure 3 plants-13-00957-f003:**
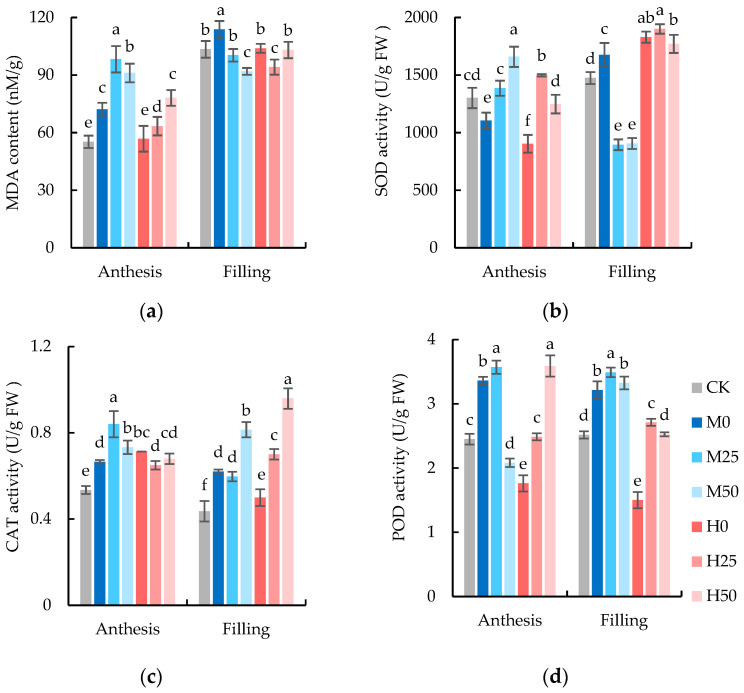
Malondialdehyde (MDA) content (**a**) and superoxide dismutase (SOD) (**b**), catalase (CAT) (**c**) and peroxidase (POD) (**d**) enzyme activities in flag leaves of wheat at anthesis and grain-filling stages. The data are expressed as the mean ± standard error (*n* = 3); different lower-case letters above each column show significance difference among treatments. CK refers to no drought stress and no proline application, M refers to 60% field water capacity, H refers to 45% field water capacity; 0, 25, and 50 denote 0, 25 mM, and 50 mM of proline solution applied in this study, respectively.

**Figure 4 plants-13-00957-f004:**
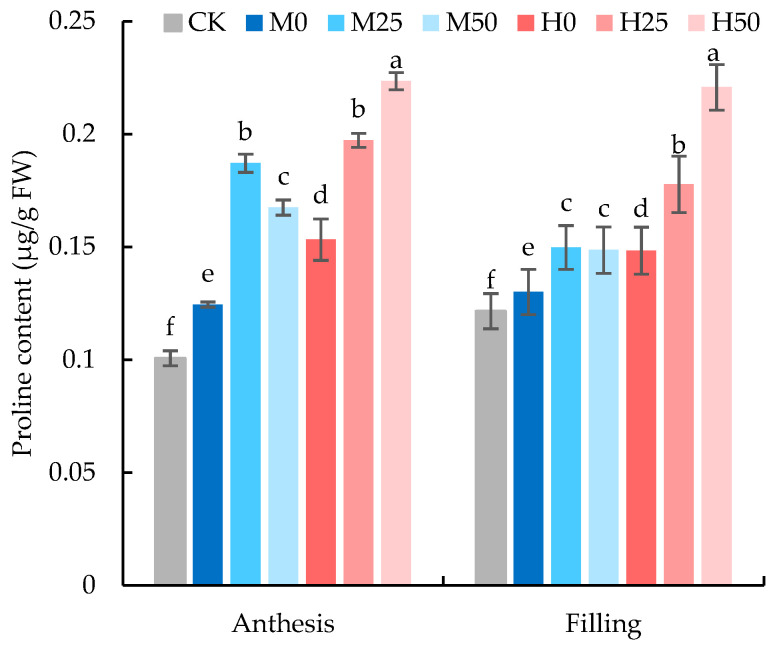
Proline content in flag leaves of wheat at anthesis and grain-filling stages. The data are expressed as the mean ± standard error (*n* = 3); different lower-case letters above each column show significance difference among treatments. CK refers to no drought stress and no proline application, M refers to 60% field water capacity, H refers to 45% field water capacity; 0, 25, and 50 denote 0, 25 mM, and 50 mM of proline solution applied in this study, respectively.

**Figure 5 plants-13-00957-f005:**
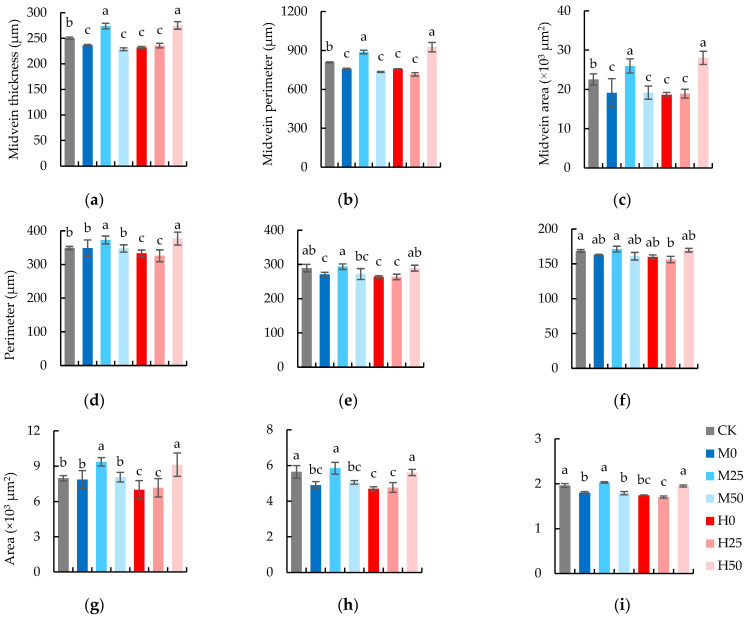
Anatomical characteristics of flag leaf: (**a**) main vein thickness, (**b**) midvein perimeter, (**c**) midvein area, (**d**) perimeter of the vascular bundle of the main vein, (**e**) perimeter of large vascular bundle, (**f**) perimeter of small vascular bundle, (**g**) area of the vascular bundle of the main vein, (**h**) area of large vascular bundle, (**i**) area of small vascular bundle. The data are expressed as the mean ± standard error (*n* = 3); different lower-case letters above each column show significance difference among treatments. CK refers to no drought stress and no proline application, M refers to 60% field water capacity, H refers to 45% field water capacity; 0, 25, and 50 denote 0, 25 mM, and 50 mM of proline solution applied in this study, respectively.

**Figure 6 plants-13-00957-f006:**
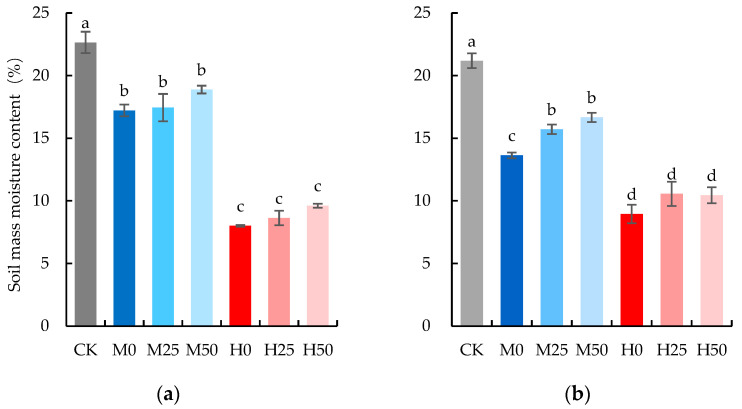
Soil mass moisture content: (**a**) 21 April 2022. (**b**) 25 April 2022. The data are expressed as the mean ± standard error (*n* = 3); different lower-case letters above each column show significance difference among treatments. CK refers to no drought stress and no proline application, M refers to 60% field water capacity, H refers to 45% field water capacity; 0, 25, and 50 denote 0, 25 mM, and 50 mM of proline solution applied in this study, respectively.

**Table 1 plants-13-00957-t001:** Two-way ANOVA results for water-holding capacity of isolated flag leaves.

Growth Stage	Source of Variation	Saturated Water Content (%)		Relative Water-Content Loss Rate (%/h)	
		F	*p*	F	*p*
Anthesis	Drought degree	5.04	0.0102 *	1.16	0.3818
	Proline concentration	12.17	0.0045 **	1.04	0.3278
	Interaction	2.5	0.1235	0.04	0.9605
Grain Filling	Drought degree	10	0.0006 ***	3.65	0.0308 *
	Proline concentration	18.36	0.0011 **	1.43	0.2554
	Interaction	15.72	0.0004 ***	7.99	0.0062 **

Note: Significant differences were observed at *p* < 0.05 *, *p* < 0.01 **, and *p* < 0.001 ***.

**Table 2 plants-13-00957-t002:** Two-way ANOVA results for SPAD values.

Growth Stage	Source of Variation	SPAD	
		F	*p*
Anthesis	Drought degree	7.09	0.0136 *
	Proline concentration	0.81	0.4561
	Interaction	0.64	0.5357
Seed formation	Drought degree	21.42	0.0001 ***
	Proline concentration	1.94	0.1656
	Interaction	2.17	0.1363

Note: significant differences were observed at *p* < 0.05 *, and *p* <0.001 ***.

**Table 3 plants-13-00957-t003:** Effect of exogenous proline on SPAD value of wheat flag leaves.

Treatment	Heading Stage	Anthesis Stage	Seed-Formation Stage
CK	46.22 ± 1.14 a	49.79 ± 1.31 a	50.43 ± 0.78 a
M0	44.70 ± 0.87 a	43.33 ± 0.39 ab	43.09 ± 0.60 b
M25	44.71 ± 1.07 a	49.00 ± 3.91 a	47.68 ± 0.84 a
M50	46.43 ± 1.11 a	46.61 ± 2.08 ab	43.38 ± 1.58 b
H0	45.47 ± 1.10 a	39.98 ± 0.70 b	39.80 ± 0.32 b
H25	45.17 ± 1.06 a	40.16 ± 0.83 b	39.82 ± 1.06 b
H50	43.55 ± 1.25 a	42.11 ± 0.94 ab	40.12 ± 0.76 b

Note: The data are expressed as the mean ± standard error based (*n* = 5); different lower-case letters in each column show significance difference among treatments. CK refers to no drought stress and no proline application, M refers to 60% field water capacity, H refers to 45% field water capacity; 0, 25, and 50 denote 0, 25 mM, and 50 mM of proline solution applied in this study, respectively.

**Table 4 plants-13-00957-t004:** Two-way ANOVA results for physical and chemical indicators of wheat flag leaves.

Growth Stage	Source of Variation	MDA Content		SOD Activity		CAT Activity		POD Activity		Proline Content	
		F	*p*	F	*p*	F	*p*	F	*p*	F	*p*
Anthesis	Drought degree	139.49	<0.001 ***	45.77	<0.001 ***	39.73	<0.001 ***	487.09	<0.001 ***	287.55	<0.001 ***
	Proline concentration	47.82	<0.001 ***	144.01	<0.001 ***	9.9	0.0029 **	174.17	<0.001 ***	220.75	<0.001 ***
	Interaction	15.13	0.0005 **	38.29	<0.001 ***	44.23	<0.001 ***	83.54	<0.001 ***	187.21	<0.001 ***
Grain Filling	Drought degree	1.87	0.1961	1035.49	<0.001 ***	14.62	<0.001 ***	1081.94	<0.001 ***	1.78	0.2072
	Proline concentration	40.52	<0.001 ***	151.39	<0.001 ***	310.8	0.0024 **	181.29	<0.001 ***	13.87	0.0008 ***
	Interaction	28.92	<0.001 ***	157.19	<0.001 ***	55.79	<0.001 ***	85.26	<0.001 ***	12.44	0.0012 **

Note: significant differences were observed at *p* < 0.01 **, and *p* < 0.001 ***.

**Table 5 plants-13-00957-t005:** Two-way ANOVA results for anatomical characteristics of flag-leaf midvein, and large and small vascular bundles.

Source of Variation	Midvein						Vascular Bundle of The Midvein			
	Thickness		Perimeter		Area		Perimeter		Area	
	F	*p*	F	*p*	F	*p*	F	*p*	F	*p*
Drought degree	0.18	0.6791	0.25	0.6239	0.35	0.5661	17.8	0.0012 **	14.94	0.0022 **
Proline concentration	16.12	0.0004 ***	10.12	0.0027 **	15.46	0.0005 ***	22.36	<0.0001 ***	15.8	0.0004 ***
Interaction	59.05	<0.0001 ***	63.57	<0.0001 ***	39.84	<0.0001 ***	69.83	<0.0001 ***	29.67	<0.0001 ***

Note: significant differences were observed at *p* < 0.01 **, and *p* < 0.001 ***.

**Table 6 plants-13-00957-t006:** Two-way ANOVA results for anatomical characteristics of flag-leaf large and small vascular bundles.

Source of Variation	Large Vascular Bundle				Small Vascular Bundle			
	Perimeter		Area		Perimeter		Area	
	F	*p*	F	*p*	F	*p*	F	*p*
Drought degree	0.71	0.4166	0.67	0.429	1.27	0.2824	1.32	0.2728
Proline concentration	1.15	0.35	1.3	0.3082	1.67	0.229	1.95	0.1849
Interaction	3.05	0.0851	3.17	0.0782	3.61	0.0594	4.64	0.0321 *

Note: significant differences were observed at *p* < 0.05 *.

**Table 7 plants-13-00957-t007:** Two-way ANOVA results for yield and the yield composition of wheat.

Source of Variation	Ear Number		Grains Number Per Ear		Thousand-Grain Weight (g)		Yield	
	F	*p*	F	*p*	F	*p*	F	*p*
Drought degree	214.05	<0.0001 ***	5.69	0.0257 *	30.3	<0.0001 ***	85.32	<0.0001 ***
Proline concentration	3.59	0.0438	0.27	0.7674	2.62	0.0945	11.95	0.0003 **
Interaction	1.63	0.2187	0.2	0.82	3.09	0.0645	8.63	0.0016 **

Note: significant differences were observed at *p* < 0.05 *, *p* < 0.01 **, and *p* < 0.001 ***.

**Table 8 plants-13-00957-t008:** Effects of exogenous proline on yield and the yield composition of wheat under drought stress.

Treatments	Ear Number	Grains Number Per Ear	Thousand Grain Weight (g)	Yield (g/barrel)	Yield (t/ha)
CK	51.60 ± 0.66 a	40.02 ± 2.17 a	37.69 ± 38.11 a	40.46 ± 0.77 a	5.36 ± 0.10 a
M0	46.40 ± 0.66 b	38.12 ± 2.13 ab	34.49 ± 31.78 ab	35.72 ± 5.31 bc	4.73 ± 0.70 bc
M25	49.40 ± 0.32 ab	39.07 ± 1.50 ab	36.08 ± 15.40 a	39.58 ± 1.15 ab	5.25 ± 0.15 ab
M50	48.00 ± 1.15 b	39.19 ± 2.60 ab	34.41 ± 71.87 ab	36.75 ± 1.02 ab	4.87 ± 0.14 ab
H0	33.20 ± 1.25 d	37.00 ± 0.92 b	26.42 ± 23.47 d	20.94 ± 0.53 e	2.78 ± 0.07 e
H25	35.60 ± 2.50 cd	37.08 ± 1.25 b	29.28 ± 28.41 cd	28.04 ± 0.71 d	3.72 ± 0.09 d
H50	37.00 ± 0.71 c	37.16 ± 1.14 b	32.07 ± 19.87 bc	32.20 ± 0.85 c	4.27 ± 0.11 c

Note: the data are expressed as the mean ± standard error (*n* = 5); different lower-case letters in each column show significance difference among treatments. CK refers to no drought stress and no proline application, M refers to 60% field water capacity, H refers to 45% field water capacity; 0, 25, and 50 denote 0, 25 mM, and 50 mM of proline solution applied in this study, respectively.

**Table 9 plants-13-00957-t009:** The basic properties of field soil at the experiment site.

Soil Layer (cm)	pH	EC (μS/cm)	AP (mg/kg)	TK (g/kg)	AN (mg/kg)	OM (g/kg)	TP (g/kg)
0–10	8.43	443.00	81.23	10.70	71.16	10.16	0.11
10–20	8.83	174.10	177.69	11.30	29.79	17.11	0.60
20–30	8.86	185.30	127.37	11.60	28.13	13.33	0.32

Note: EC refers to electric conductivity, AP refers to available potassium, TK refers to total potassium, AN refers to alkali-hydrolyzed nitrogen, OM refers to organic matter, TP refers to total phosphorus.

## Data Availability

The data presented in this study are available on request from the corresponding author.

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
