# Peer review of "Proline Spray Relieves the Adverse Effects of Drought on Wheat Flag Leaf Function"

_plants, 2024, doi:10.3390/plants13070957_

Round 1

Reviewer 1 Report

Comments and Suggestions for Authors

Dear Editor, many thanks for your kind invitation to evaluate the MS, entitled: “Effect of spraying proline on regulating flag leaf function and yield of wheat (Triticum aestivum L.) under drought stress”. I am sure this study will be useful for understanding the physiological responses of wheat toward the performances of yield and economic return among different proline and drought stress configurations. However, before considering publishing in the journal, there is a minor issue is required for this article as follows:

1.      Authors have to clarify the novelty of their study in the abstract.  

2.      Keywords should be different than the title to increase the visibility of your work

3.      The authors need to improve in introduction and discussion by creating connections around the area of other reports on the regulation of biomass allocation ratio and water use efficiency in wheat plants is also a basic responsive mechanism to drought stress and therefore the authors are recommended to add different statement on wheat responsive mechanism to different abiotic stress,

Therefore, please before the sentence "In recent years, the frequency,…………." Add the following statement " Over the past decade, most research in the agronomy section has emphasized understanding the effect of such deviations in abiotic stresses on wheat plant growth and production, which can cause stress such as heat (Liu et al., 2020), salt, drought and water deficit (Gui et al. 2020).

-          Liu et al. (2020) TaHsfA2-1, a new gene for thermotolerance in wheat seedlings: characterization and functional roles. Journal of Plant Physiology 246: 153135.

-          Gui et al. (2020). Differentiate effects of non-hydraulic and hydraulic root signaling on yield and water use efficiency in diploid and tetraploid wheat under drought stress. Environmental and Experimental Botany 181, 104287. 

4.      Please authors should improve the hypothesis of the study and use the different parameters to build a solid story related to the related mechanism of proline to regulate the drought stress on wheat plant.  

5.      The results section is very confusing because of the language. It is hard to understand in many places.

Please accept this paper shall the author address the above mentioned minor points

Comments on the Quality of English Language

No specific comments

Author Response

Response to Reviewer

Dear Reviewer/Editor,

We are grateful for the time and effort the reviewers have taken to help us improve the manuscript. Our responses to Reviewer 1 are marked in green in the revised manuscript.

Reviewer 1

However, before considering publishing in the journal, there is a minor issue is required for this article as follows:

  1. Authors have to clarify the novelty of their study in the abstract.

Reply: The novelty of the study has been clarified in the abstract. Line 11-12.

  1. Keywords should be different than the title to increase the visibility of your work

Reply: We revised the keywords. Line 25.

  1. The authors need to improve in introduction and discussion by creating connections around the area of other reports on the regulation of biomass allocation ratio and water use efficiency in wheat plants is also a basic responsive mechanism to drought stress and therefore the authors are recommended to add different statement on wheat responsive mechanism to different abiotic stress.

Reply: Thank you for your suggestion. We included relevant statements in the introduction and discussion. Line 39-40, 43-44 and 381-382.

  1. Therefore, please before the sentence "In recent years, the frequency,…………." Add the following statement " Over the past decade, most research in the agronomy section has emphasized understanding the effect of such deviations in abiotic stresses on wheat plant growth and production, which can cause stress such as heat (Liu et al., 2020), salt, drought and water deficit (Gui et al. 2020).

Liu et al. (2020) TaHsfA2-1, a new gene for thermotolerance in wheat seedlings: characterization and functional roles. Journal of Plant Physiology 246: 153135.

Gui et al. (2020). Differentiate effects of non-hydraulic and hydraulic root signaling on yield and water use efficiency in diploid and tetraploid wheat under drought stress. Environmental and Experimental Botany 181, 104287.

Reply: We add the statement as you suggested. Line 34-37.

  1. Please authors should improve the hypothesis of the study and use the different parameters to build a solid story related to the related mechanism of proline to regulate the drought stress on wheat plant.

Reply: We have added a sentence in Introduction to refine the hypothesis of the study. Line 94-98.

  1. The results section is very confusing because of the language. It is hard to understand in many places.

Reply: Thank you for your suggestion. We have polished the language in the results section in the revised version. Line 109-110, 125, 129, 139-140, 143-145, 152-153, 157-159, 218-219, 238-239, and 255-256.

Reviewer 2 Report

Comments and Suggestions for Authors

Review on “Effect of spraying proline on regulating flag leaf function and yield of wheat (Triticum aestivum L.) under drought stress”. The authors examined the effect of severe and mild drought stress on wheat and the application of different proline concentrations against drought stress. They stated that severe drought had more significant effects on the measured parameters like leaf saturated water content, water holding capacity, SPAD value, leaf anatomical characteristics, and reduced yield. I have to say these results are not considered new results in drought stress physiology examinations. However, the application of proline as a foliar treatment mitigated the negative effects of drought stress. The authors suggest using 25 mM proline foliar treatment under mild and 50 mM proline treatment under severe drought conditions.

Generally, I think that the topic of the paper is very interesting and has great importance. Overall the manuscript is clear, organized, and well-structured. The introduction was well-written and gave a good background of the main idea of the study. Results are statistically analyzed. The manuscript contains many valuable results.  However, the manuscript needs some minor improvements and corrections. Please check the Oxford comma throughout the manuscript.

Abstract:

Line 9. Please correct: Drought stress is one of the key factors…

Keywords:

Please arrange the keywords in alphabetical order.

Introduction:

Would you please add some information regarding the wheat production in China? (e.g. growing area, average yield, how many t/year production, etc.). Thank you.

Please add the hypothesis of the study to the last paragraph. (conduction of an experiment is not a hypothesis)

Results:

Please add the number of repetitions of each measured parameter under each figure/table.

Materials and Methods:

Figure 6. There are no CK, M, H, etc. presented on the diagram.

Conclutions:

Please add information related to the future. Like: would it be possible to use proline against drought stress under field crop production in the future?

Author Response

Response to Reviewer

Dear Reviewer/Editor,

We are grateful for the time and effort the reviewers have taken to help us improve the manuscript. Our responses to Reviewer 2 are marked in red in the revised manuscript.

Reviewer 2:

Please check the Oxford comma throughout the manuscript.

Reply: Done.

Abstract:

Line 9. Please correct: Drought stress is one of the key factors…

Reply: Done. Line 9.

Keywords:

Please arrange the keywords in alphabetical order.

Reply: Done.

Introduction:

Would you please add some information regarding the wheat production in China? (e.g. growing area, average yield, how many t/year production, etc.). Thank you.

Reply: We added the above information in the introduction. Line 29-32.

Please add the hypothesis of the study to the last paragraph. (conduction of an experiment is not a hypothesis)

Reply: We added. Line 94-98.

Results:

Please add the number of repetitions of each measured parameter under each figure/table.

Reply: Done.

Materials and Methods:

Figure 6. There are no CK, M, H, etc. presented on the diagram.

Reply: Done.

Conclutions:

Please add information related to the future. Like: would it be possible to use proline against drought stress under field crop production in the future?

Reply: Thank you for your suggestion. We added the information in the conclusion. Line 506-508.

Reviewer 3 Report

Comments and Suggestions for Authors

The study is devoted to the important issue of alleviating the effects of drought stress on wheat growth through foliar application of proline. The authors carefully present experimental data that contribute to the understanding of the influence of proline on the activity of antioxidant enzymes, the content of osmotic regulatory substances and the tissue structure of wheat flag leaves. The manuscript contains several mistakes and unclear sentences that need to be explained and corrected and some data added.

Author Response

Response to Reviewer

Dear Reviewer/Editor,

We are grateful for the time and effort the reviewer have taken to help us improve the manuscript. Our responses to Reviewer 3 are marked in blue in the revised manuscript.

Reviewer 3:

The title seems a bit confusing, consider rephrasing/rewording.

Reply: We revised as suggested.

Abstract

L11-12 Check and rephrase “….a barrel experiment grown with wheat…”

Reply: We rephrased the sentence. Line 12-15.

L18-19 Unclear, check the meaning “…improved the relationship between flag leaves and water….”

Reply: We revised. Line 19-22.

L21-22 Presenting one value does not make much sense, it would be more appropriate to show the improvements in yield by proline application for all variants

Reply: We added more data. Line 22-24.

Keywords

Use other words than the title contains

Reply: We revised the keywords as you suggested. Line 27.

Introduction

L29 Here and elsewhere, use relevant citations or do not use them. Here you confirm that water is an important environmental condition by citing the article on lentils.

Reply: We removed that reference.

L34-36 (also 41-43) This information makes no sense without comparing what level of soil water content the increase or decrease was compared to.

Reply: We added the soil water level as you suggested. Line 44-45, 54.

L36 Here and elsewhere unify the use of spaces at the end of the sentence in the text

Reply: We checked the spaces and unified.

L70 Why you write the Latin names for common species, you should rather give them for less common species

Reply: We revised it. Line 82.

Results

The results are described and presented mostly clearly, to a reasonable extent, graphs are well-readable.

Methods

L90 Describe in Methods, how “The SMC daily variation” or “daily average change” was calculated. The term variation is confusing, consider using water consumption, (rate of) evapotranspiration

Reply: We revised them to “evapotranspiration”, and rewrite this section. Line 105-114.

L154 In material and methods you mentioned the date 4 May 2022, there you write 11 May 2022. What is correct?

Reply: 11 May 2022 is correct. We corrected the error in Line 470.

L234-235 “the yield composition “It is a little bit clumsy expression, please rephrase it

Reply: Based on a literature survey, “the yield composition “is a common expression, so we keep using this expression.

Discussion

The discussion is of reasonable length, with relevant citations, and experimental data is interpreted correctly.

L257 how do you mean?

“the water transport of wheat flag leaf” in flag leaf, through flag leaf?

Reply: We revised as you suggested. Line 289.

L286 “prohibited” Is it true? Decreased, reduced...

Reply: Revised to inhibited. Line317.

Materials and Methods

Both the experiment and the methods are well described. There are some inexactness or topics for additions.

L358-9 Add the standard name of soil type

Reply: We checked and corrected this statement. Line 391-392.

L363 Add the value of soil wilting point; it enables to calculate available water capacity

Reply: Soil wilting point was added in Line 396-397.

L364 It is better to express the values in kg/ha

Reply: We revised as you suggested. Line 397-398.

L387 “…soil moisture was controlled as mentioned above” Unclear, do you mean soil moisture was determined several times in the period? How many times?

Reply: We measured the soil mass moisture content 3 times in the period. Line422.

Add soil moisture at the end of the period (27 April) to the graph as you show only a daily variability of SMC in Results (Fig. 1).

Reply: The drought stress lasted until the evening of 27 April, then all the treatments were watered (rehydration). We did not measure the soil moisture content at the end of drought stress (the evening of 27 April), hence we added soil moisture on 25 April 2022 (Figure 6b) considering that 25 April was the last time we measured the soil mass moisture content. Line 430.

L388 Add information on how the “rehydration”

(L121) of treatments was performed. Was SMC monitored during grain filling?

Reply: From the evening of 27 April to 2 days before harvest, all the treatments were irrigated with 2 L of water every 2 or 3 days according to weather condition to fully meet the water needs of wheat growth. We did not monitor SMC during grain filling. Line 427-429.

Figure 6 Correct, but add X-axis description

Reply: We corrected Figure 6.

L408 Add reference to the method of determining RWC as commonly it is based on leaf weight before and after soaking with water, your approach is different

Reply: I agree with you that REC is based on leaf weight before and after soaking with water commonly. In that method, the sheaths need to be removed from the leaves which may affect the ability of water uptake. In this paper, the complete leaves with the sheaths were employed to absorb water till the leaves were saturated, and calculated the RWC based on saturated moisture content of leaves which is a more easy-to-control uniform state. In this way, RWC of all treatments were projected between 0 (completely dry state) and 1 (saturated state), which is easier to be compared. We added the reference in Line 458.

L409; 419 Use the standard phenological growth description of the stages (BBCH)

Reply: We added BBCH growing stage as you suggested. Line 455, 456, 470, and 471.

L444 No results of ANOVA are presented or commented on, only results of the Duncan test; I suggest adding the significance of water and proline application treatments on monitored characteristics

Reply: We added Two-way ANOVA analysis to identify the effects of soil drought degree and proline concentration, and these results were presented by five tables in the Supplementary Material. Line 493, 495, and the supplementary material.

Round 2

Reviewer 3 Report

Comments and Suggestions for Authors

Thanks to the authors for the answers and additions.

And from my point of view, I consider the article fit for publication.

I only have a small comment on line 31-32. It would be better to state the average yield in t/ha.

Author Response

Dear reviewer,

 Thank you for your suggestion. We add the average yield in t/ha to Table 7 and marked in red in the revised manuscript.